# Feasibility and Acceptability of a Remotely Delivered Executive Function Intervention That Combines Computerized Cognitive Training and Metacognitive Strategy Training in Chronic Stroke

**DOI:** 10.3390/ijerph20095714

**Published:** 2023-05-04

**Authors:** Abhishek Jaywant, Leah Mautner, Rachel Waldman, Michael W. O’Dell, Faith M. Gunning, Joan Toglia

**Affiliations:** 1Departments of Psychiatry & Rehabilitation Medicine, Weill Cornell Medicine, New York, NY 10065, USA; 2Department of Rehabilitation Medicine, Weill Cornell Medicine, New York, NY 10065, USA; 3Department of Psychiatry, Weill Cornell Medicine, New York, NY 10065, USA; 4School of Health and Natural Sciences, Mercy College, Dobbs Ferry, NY 10522, USA

**Keywords:** stroke recovery, cognitive rehabilitation, telerehabilitation, executive function

## Abstract

Executive dysfunction after stroke is associated with limitations in daily activities and disability. Existing interventions for executive dysfunction show inconsistent transfer to everyday activities and require frequent clinic visits that can be difficult for patients with chronic mobility challenges to access. To address this barrier, we developed a telehealth-based executive function intervention that combines computerized cognitive training and metacognitive strategy. The goal of this study was to describe intervention development and to provide preliminary evidence of feasibility and acceptability in three individuals who completed the treatment protocol. The three study participants were living in the community and had experienced a stroke >6 months prior. We assessed satisfaction (Client Satisfaction Questionnaire-8 [CSQ-8]), credibility (Credibility and Expectancy Questionnaire), and feasibility (percent of sessions completed). All three subjects rated the treatment in the highest satisfaction category on the CSQ-8, found the treatment to be credible, and expected improvement. Participants completed a median of 96% of computerized cognitive training sessions and 100% of telehealth-delivered metacognitive strategy training sessions. Individuals with chronic stroke may find a remotely delivered intervention that combines computerized cognitive training and metacognitive strategy training to be feasible and acceptable. Further evaluation with larger samples in controlled trials is warranted.

## 1. Introduction

Cognitive dysfunction is a major contributor to disability after stroke. Among cognitive abilities, deficits in working memory and executive functions are especially common and persistent [1] and closely associated with limitations in activities of daily living [2,3]. Post-stroke executive dysfunction is thought to emerge from local and global disruption to white matter tracts and functional brain networks including the executive control network, default mode network, and salience network [4,5].

Among existing interventions, computerized cognitive training (CCT) and metacognitive strategy training (MST) are two popular but contrasting approaches that differ in key treatment ingredients and targets. CCT uses a bottom-up, restorative approach that focuses on improving the underlying cognitive impairment through repetitive practice of cognitive exercises that gradually place greater demands on cognitive skills. CCT exercises typically adjust to the person’s ability so that training is always delivered at the “just right level” of challenge or at the edge of the person’s ability level. The repetition, novelty, immediate feedback, and continuous adjustment of task difficulty based on the person’s performance are thought to be critical in promoting neuroplasticity [6]. Findings on CCT for working memory and executive function in stroke have been mixed [7,8]. The literature suggests near transfer effects to similar tasks and modulation of brain networks underlying executive dysfunction, but more limited evidence for generalization to everyday life activities. Based on these findings, the American Congress of Rehabilitation Medicine Cognitive Rehabilitation Taskforce cautioned against the standalone use of CCT [9].

In contrast, MST uses a top-down approach that focuses on training a person to monitor and evaluate their cognitive performance, typically within the context of functional or everyday activities. The person is guided in identifying challenges and generating strategies to manage cognitive performance difficulties. Key ingredients include a focus on strategies, self-monitoring skills, use of guided metacognitive questions before and/or after activities, and methods for explicitly bridging strategies to other life activities [10]. Recent studies support the benefits of MST within occupation-based treatment for promoting both functional improvement and transfer of functional skills after stroke [11,12,13].

Despite both approaches being grounded in different theoretical bases, there are potential advantages to their integrated use. While CCT approaches typically target specific cognitive abilities, MST targets broader skills such as self-awareness, self-monitoring skills, and personal strategies that might be effective in promoting generalization to everyday function. It may be that the simultaneous use of both methods is more effective than each alone because of their complementary nature. There is also some evidence to suggest that CCT and MST might modulate similar underlying neural circuits, including the executive control network and default mode network [14,15]. The combination of CCT and MST has demonstrated efficacy in persons with schizophrenia and in children with acquired brain injury [16,17], but has yet to be adequately examined in adults with stroke. In addition, there is a lack of information describing the process taken to integrate these two approaches in a way that is feasible and accessible.

A barrier for individuals with stroke when accessing these interventions is their intensive nature, which requires frequent in-person clinic visits. The frequency and intensity of CCT is associated with treatment gains and recent interventions tested in stroke have required in-person visits 5–7 days/week [18,19]. MST also typically requires several in-person sessions with a clinician to facilitate self-awareness, cognitive strategy use, and generalization. Mobility and ambulation challenges are highly prevalent after stroke [20,21,22], and their presence may impact the ability of individuals with stroke to travel to in-person appointments. This barrier may also widen existing disparities in care as persons with stroke from marginalized groups and of lower socioeconomic status may find it even more difficult to access treatment. With the ubiquity of telemedicine and virtual care in the context of the COVID-19 pandemic, persons with stroke may in fact prefer the ease of conducting remote CCT notwithstanding the above access barriers.

Emerging evidence suggests that virtually delivered neurorehabilitation may be feasible and efficacious after stroke [23]. A large randomized controlled trial has shown that remotely delivered motor rehabilitation is feasible and associated with comparable gains to in-person rehabilitation in [24]. A six-week telehealth-delivered memory rehabilitation program for stroke patients was found to be feasible and associated with subjective memory improvements [25]. To date, however, there is limited evidence on whether a remotely delivered executive functioning intervention, particularly one that combines CCT and MST, is feasible or acceptable in chronic stroke. Studying the feasibility of a remotely delivered CCT + MST intervention for executive dysfunction is important given the frequent disturbance in executive functions after stroke and the association between executive functions and activities of daily living.

The present study had two goals: (1) to describe the development of a combined executive function intervention in chronic stroke that integrated CCT with MST; and (2) to provide preliminary evidence of the feasibility and patient engagement of this intervention in a remotely delivered format. We also explored pre-treatment to post-treatment clinical changes in neuropsychological assessments, patient reported outcome measures, and performance-based assessment of cognitive-instrumental activities of daily living (C-IADL).

## 2. Materials and Methods

### 2.1. Participants

In this initial intervention development and pilot phase, we recruited three community-dwelling individuals in 2019–2020 from the New York City area through referrals by clinicians and through contacting individuals on an internal hospital stroke registry. Inclusion criteria included a history of first-time stroke (minimum of six months prior to enrollment), English speaking, ability to comprehend sufficiently to participate in the intervention, subjective or objective evidence of cognitive difficulties (as determined by the telephone MoCA or self-report), willingness to participate in full study duration, ability to operate a computer keyboard and mouse, not concurrently receiving other cognitive rehabilitation services, and cognitively able enough to perform basic self-care activities. Exclusion criteria included a history of other neurologic disorder, history of severe mental illness or alcohol/substance use disorder, current severe depression requiring referral for psychiatric care, or history of dementia or dependence in basic self-care activities due to cognitive deficits. All procedures were approved by the Weill Cornell Medicine Institutional Review Board. All study participants provided written informed consent. The study was registered on ClinicalTrials.gov (Identifier: NCT04098835).

### 2.2. Intervention Development

To guide intervention development, we incorporated the UK Medical Research Council framework for developing and evaluating complex interventions [26] and specifically, consensus guidelines on how to develop complex interventions to improve health as articulated by O’Cathain et al. [27]. Table 1 provides each of the components of O’Cathain’s framework for intervention development, how they were incorporated in the current study, key findings, and next steps/follow-up questions.

Table 2 summarizes our intervention using the Template for Intervention Description and Replication (TIDieR) checklist. As noted in the Introduction, we sought to develop an intervention that integrated CCT and MST to target executive functions and explicitly train for transfer to everyday C-IADLs. We chose Rehacom (Hasomed, GmBH) as the cognitive training software given evidence for efficacy in stroke [19,28], multiple training tasks focused on executive functions, and training tasks that visually resembled everyday C-IADLs (e.g., using playing cards, performing mental calculations using money/currency, dividing attention in a simulated driving environment). For the MST component, we used the Multicontext approach [10,29]. The goal of the Multicontext approach is to help the participant increase their awareness of cognitive performance and to self-generate and use cognitive strategies to manage cognitive lapses. Strategies are practiced along a “horizontal continuum” to promote transfer and generalizability—that is, the same strategy with similar cognitive demands is practiced across multiple everyday life situations. This generalizability is important, as many individuals with stroke often demonstrate difficulty connecting similarities across experiences—including similar cognitive skills hindering performance. The Multicontext approach uses a key component of MST, which is guided questions before and after activity experiences.

Our combined intervention comprised 25 computerized cognitive training sessions using the Rehacom software (Hasomed GMBH, Magdeburg, Germany), each 30 min in duration; 8 remotely delivered MST sessions with a neuropsychologist using the Multicontext approach; and a workbook of homework activities based on the Multicontext approach completed independently by the participants. Twenty-three out of 25 CCT sessions were conducted independently by the participants at home on a loaned computer with preloaded software; 6/8 MST coaching sessions were conducted by Zoom teleconference, and all workbook activities were completed by the participant independently at home. Thus, most of the treatment was implemented remotely with most of the clinician–participant interaction occurring via videoconference. We elected to have two sessions each of CCT and MST completed in-person—one at the beginning of treatment and one in the middle of the study—to provide opportunities for troubleshooting any difficulties, navigating barriers, and ensuring adherence to the intervention.

Participants completed three Rehacom exercises daily, each approximately 10 min in duration. CCT exercises were structured in a “bottom-up” fashion such that initial training activities focused on lower-level attentional functions and subsequently progressed to higher order executive functions (e.g., dividing attention/dual tasking, planning, organizing, and problem-solving). The initial lower-level attentional training targeted reaction time, attention to detail, and processing speed. Higher-order executive function exercises included organizing a shopping list and shopping in a grocery store, dividing attention in a driving game, and mental manipulation using currency/money. Because we conceptualized working memory as a component of executive function and a core dysfunction, 10 min of a working memory exercise was used every day throughout the intervention. This exercise requires the participant to maintain in mind, sequence, and manipulate an increasing amount of information using stimuli that are stylized as playing cards. All CCT exercises were adaptive and changed in difficulty based on the participant’s performance. Duration and frequency of training were chosen following a complete review of existing Rehacom studies.

Before and after Rehacom exercises, participants were asked to answer questions based on the Multicontext approach. These questions were designed to help the participant anticipate challenges, generate strategies, and observe links between Rehacom exercises and everyday C-IADLs. Participants wrote answers to these questions in a printed workbook that was provided at the beginning of treatment. Before each Rehacom session, participants were asked questions such as “what challenges do you anticipate on today’s exercises?” and “what strategies, tricks, or special methods could you use to keep track of the information?” After the Rehacom exercises, participants answered questions such as “what challenges did you run into?” and “what strategies did you use?” and “what could you do differently next time?” Participants were asked to write down how the challenges encountered, and strategies used on Rehacom exercises, related to their everyday C-IADLs.

In addition to CCT with accompanying guided questions, participants completed workbook activities that required keeping track of information across different functional activities (e.g., keeping track of 5 items on a shopping list, on a calendar, TV schedule, etc.). Participants answered similar questions before and after the functional tasks regarding anticipated challenges, potential strategies, encountered challenges, and strategies used. To assist participants in making links between functional activities, CCT, and everyday C-IADLs, specific questions were used to probe how similar challenges could occur and strategies could be used across these different contexts.

Finally, the MST coaching sessions used a similar approach in which participants completed a structured activity in the workbook with a clinician (study author AJ, a clinical neuropsychologist). Prior to the activity, the clinician asked guided questions to help the participant anticipate challenges and generate strategies. After the task, the clinician asked further questions to help the client reflect on challenges and strategies, and to bridge the activities to the Rehacom sessions and to everyday C-IADLs. Coaching sessions also included problem-solving barriers to engagement in the treatment and discussion (and positive reinforcement) of the participant’s progress. The last coaching session focused on reviewing and consolidating gains and setting goals for the future.

### 2.3. Assessments

#### 2.3.1. Client Satisfaction Questionnaire-8 (CSQ-8)

The CSQ-8 [30] was the primary measure of acceptability and satisfaction. The CSQ-8 includes 8 questions that are rated on a 1–4 Likert-type scale where higher scores indicate greater acceptability and satisfaction. Based on standard convention, we classified each score as low satisfaction (score of 8–20), moderate satisfaction (score of 21–26), and high satisfaction (score of score of 27–32). The CSQ-8 score at the end of treatment was the primary outcome of interest. We also administered the CSQ-8 at the midpoint of the intervention to explore participant satisfaction with the initial portion of the treatment, particularly because treatment adherence/dropout can be a concern in complex interventions such as ours.

#### 2.3.2. Credibility and Expectancy Questionnaire (CEQ)

The CEQ [31] was administered to evaluate how believable and logical the participant perceives the treatment to be, as well as their expectancy of change. The first three items ask, on a 9-point Likert-type rating scale, how logical the intervention is perceived to be, how successful the participant thinks the intervention will be in raising quality of functioning, and how confident the participant would be in recommending the intervention to a friend with similar problems. The CEQ Credibility score is calculated as the average of these three items. The CEQ Expectancy score is determined using a separate item which asks the participant to rate “how much improvement in your functioning do you think will occur” by the end of the intervention. This question is rated on a 0–100% scale in 10 percentage point increments. The CEQ scores at the beginning of treatment was the primary outcome of interest. We also administered the CEQ at the midpoint of the intervention to explore whether participants continued to find the intervention credible, with positive expectancy of change, as this could again inform the potential for adherence and dropout.

#### 2.3.3. Feasibility

We assessed feasibility through the percent of CCT sessions completed and the percent of MST sessions completed.

#### 2.3.4. Neuropsychological Battery

To explore clinical outcomes following the intervention, we administered at baseline and end of treatment the Wechsler Adult Intelligence Scale–Fourth Edition Digit Span subtest (auditory attention and working memory); the Wechsler Memory Scale–Fourth edition Symbol Span subtest (visual attention and working memory); the Symbol–Digit Modalities Test (SDMT; processing speed and divided attention); the Trail Making Test (TMT) A and B (processing speed, working memory, and set-shifting/cognitive flexibility); the Paced Auditory Serial Addition Test (PASAT; processing speed and working memory); the Behavior Rating Inventory of Executive Function (BRIEF; self-report of executive functioning difficulties).

#### 2.3.5. Weekly Calendar Planning Activity (WCPA)

The WCPA [32] was used as a standardized and performance-based clinical measure of cognitive-instrumental activities of daily living (C-IADL). The task requires participants to organize appointments into a weekly schedule. Appointments are either fixed at a specific day and time or flexible, requiring the participant to manage conflicting appointments. The participant must keep track of multiple rules while managing distractors. Thus, the task integrates multiple executive functions including planning, problem-solving, working memory, inhibition, and flexibility. Outcome variables include the percentage of 17 total appointments entered correctly; strategies used; and rules followed.

### 2.4. Procedure

#### 2.4.1. Assessments

Prior to enrollment, a telephone screening was conducted. Screening included information on demographics and medical history as well as a telephone Montreal Cognitive Assessment (MoCA) and a self-report of cognitive functioning. Participants who performed below the clinical cutoff on the telephone MoCA or who self-reported cognitive difficulties were enrolled in the study. Enrolled participants completed baseline and end-treatment visits consisting of the study assessments described above. An additional mid-treatment in-person assessment was also conducted.

#### 2.4.2. Technology Use

As part of the initial (in-person) CCT session, participants completed training in the use of the loaned laptop computer and Rehacom software. The study clinician (AJ) demonstrated use of the hardware and software and then the participants had the opportunity to practice with corrective feedback as necessary. The extent of this training and practice was individualized for each participant. One participant required minimal training, as she routinely used computers for her occupation. Another participant required more repetitive practice, and a third participant had assistance from a caregiver to help with both hardware and software. All participants (and a caregiver if present) were given an instruction sheet with step-by-step instructions on starting and shutting down the laptop computer and logging into the Rehacom program. Participants also had the clinician’s contact information for troubleshooting needs if necessary.

#### 2.4.3. Intervention

In between these assessments, participants completed the intervention over a period of 5 weeks.

### 2.5. Statistical Analyses

Because of the small sample size, statistical analyses used descriptive statistics (frequency counts/percentages, median, range) as well as visual inspection of data to evaluate trends. Neuropsychological test scores were converted to demographically adjusted z-scores using published normative data to facilitate clinical interpretation and subsequently graphed for visual inspection.

## 3. Results

### 3.1. Demographics

All participants were >6 months post-stroke and living in the community. P1 was a single White woman in her 60s living independently after a left temporoparietal hemorrhage approximately 5 years prior. P2 was a married White man in his 50 s with a history of a left temporoparietal ischemic stroke approximately 2 years prior who received support from family due to motor limitations. P3 was a single Hispanic woman in her 40s employed full-time with a history of left hemisphere ischemic stroke in the posterior region approximately 1.5 years prior.

### 3.2. Satisfaction, Credibility, Expectancy, and Feasibility

Table 3 provides scores on the CSQ-8, CEQ, and feasibility by participant. All CSQ-8 scores fell within the high satisfaction range of 27–32, suggesting consistently high satisfaction. Based on the CEQ at baseline and mid-treatment, participants perceived the treatment to be credible and expected a positive change in function. Participants completed a high percentage of CCT sessions. All participants completed 8/8 MST coaching sessions.

### 3.3. Weekly Calendar Planning Activity

Table 4 demonstrates WCPA scores at baseline, end of treatment, and as change from baseline to end of treatment. All three participants demonstrated an increase in the number of correct responses. Two out of three participants had an increase in the number of strategies used and rules followed.

### 3.4. Individual Participant Performance

Participant P1 consistently completed workbook exercises, although they required sporadic reminders due to memory difficulties. She otherwise was motivated and engaged throughout the intervention. P1’s ability to identify and articulate challenges on CCT exercises progressed through treatment. Initially, she identified vague challenges such as “the arrows on the keyboard” and the “card game”. By the end of treatment, she described challenges on CCT more specifically such as “anything that’s spatially oriented, I have to look extra close so I don’t make a mistake”. During MST coaching sessions, strategy use began to emerge in the third session, where she attempted to categorize items in the activity. In the fourth session, double-checking and underlining details were discussed as strategies. In subsequent sessions, P1 identified as strategies (during pre-task questioning) re-reading instructions and underlining key details, as well as breaking tasks up into more manageable components. Through MST sessions and workbook exercises, P1 made links between CCT and workbook exercises and everyday C-IADLs. She noted that she organizes a stroke support group and the strategies she learned could help her to keep up with organizational tasks related to the group. At the end of treatment, P1’s WCPA (Table 4) and neuropsychological test performance (Table 5) was largely similar to baseline; certain scores (e.g., Symbol Span, TMT A) were in fact lower than baseline. It is notable that P1’s baseline assessment was conducted early in the day and her end of treatment assessment could only be completed at the end of the day at which time she reported significant fatigue. She was aware of the impact of fatigue on cognitive performance, remarking that it would be beneficial to “complete exercises in the morning”.

During the intervention, P2 was motivated and engaged; however, he required support from caregivers because of motor and expressive language deficits. P2 identified challenges in CCT and workbook exercises early in coaching sessions. The main challenge identified was remembering to keep track of large amounts of information, both in CCT and MST exercises. Strategy use also emerged early and strategies identified included removing distractions, making “mental notes”, using acronyms, organizing information into numerical or alphabetized lists, and crossing information off as it was completed. P2 made links to everyday C-IADLs such as driving and keeping a schedule. Responses to MST questions often required prompting for elaboration due to the participant’s mild chronic expressive aphasia. At the end of treatment, P2 showed improvement on the WCPA, Trail Making Test-A, Stroop Color-Word, and Paced Auditory Serial Addition Test. Strategies that this participant reported using included “making a mental note”, “moving distractions away”, and “organizing groups of items to make it easier”.

P3 was motivated and engaged in the intervention, though a barrier to participation was that she worked a full-time job. This prevented her from completing MST workbook exercises and required collaborative problem-solve to schedule MST coaching sessions in the evenings and reschedule several CCT and MST sessions to different dates and times. During MST coaching sessions, P3’s description of challenges was initially vague and task-specific (“pay attention to the suit and the number on the card game”) but, with questioning, became clearer and more generalized (i.e., realizing that she has trouble keeping track of two things at once). Strategy generation emerged early in MST coaching sessions, where the primary strategy identified and used was mental rehearsal (“say it in my head three times”). P3 generalized challenges and strategies to the administrative and organizational tasks she had to complete at work. At the end of treatment, P3 showed improved performance on the WCPA, Digit Span, Symbol Span, Stroop Word Reading, and the BRIEF.

## 4. Discussion

The goal of this study was to (1) describe the development of remotely delivered executive function intervention that combines CCT and MST, and (2) to provide preliminary evidence of feasibility and acceptability in three individuals with chronic stroke.

CCT and MST represent somewhat different approaches to cognitive rehabilitation. CCT uses a “bottom-up” approach to train specific cognitive abilities intensively and adaptively, whereas MST uses a “top-down” approach that increases individuals’ moment-to-moment awareness of cognitive performance and ability to implement cognitive strategies in real-world settings. Because CCT has shown the potential to modulate brain networks underlying executive functions, but variable evidence of transfer to everyday activities, we combined CCT with MST.

In this manuscript, we describe the development of our combined CCT + MST intervention. We conducted an extensive literature review and relied on the expertise of an interdisciplinary team of researchers and clinicians with expertise in neurorehabilitation, neuropsychology, neuroscience, MST, and scalable intervention development. The rationale for combining CCT and MST was based on theories of brain plasticity, learning, strategy acquisition and application, and generalization. We decided to use Rehacom software because of its existing evidence based in stroke and the inclusion of cognitive training games that visually represent common everyday activities such as driving and shopping, which we believed would harmonize well with function-based MST. The MST component used the Multicontext approach, which is designed to explicitly train for transfer to everyday activities. A core component of our intervention was the use of guided questions to help participants increase awareness of challenges, self-generate cognitive strategies, and to observe links between CCT games, MST exercises, and real-world activities.

The results from piloting the intervention over a 5-week period indicate that all three participants reported high satisfaction with the intervention, found it credible and had expectancy of change. All three participants completed a high percentage of CCT sessions and MST coaching sessions. Satisfaction was measured using the CSQ-8. On this instrument, all three participants reported high satisfaction (score of 27–32) at the midpoint of treatment and at the end of treatment. This finding is consistent with prior studies using telehealth-based, remotely delivered interventions for motor dysfunction and subjective memory impairment [24,25], in which individuals with stroke reported high satisfaction with remote intervention. In our study, participants benefitted from an initial in-person session to familiarize them with the technology and to build rapport, as well as a mid-treatment in-person session to help problem-solve barriers or technical difficulties. One participant required additional phone contact with the study clinician to troubleshoot technical difficulties with the CCT software. Thus, the extent to which executive function interventions such as ours can be administered fully remotely vs. having occasional in-person visits remains an important question for future research.

We additionally used the CEQ to evaluate credibility and expectancy of change. At the outset of the intervention, as well as at the mid-point, participants found the intervention to be both credible (i.e., to have “face value” as a logical intervention) and had high expectancy of change. Though the high expectancy of change (as well as our small sample) precludes evaluation of efficacy given the lack of a placebo control group, it does provide evidence that participants were motivated to complete the study. While further pilot testing on our approach is needed given the small sample, our finding of high satisfaction and feasibility is consistent with previous studies that have investigated remote intervention in chronic stroke for mobility and memory impairment [24,25].

Indeed, we found that feasibility was high. That is, participants had high rates of completion of CCT exercises and remotely delivered MST coaching sessions. Qualitatively, one participant (P3) who was still employed was only able to complete coaching sessions in the evenings, which required flexibility on the part of the research staff.

Although not the primary focus of this intervention development and preliminary pilot study, we explored changes in neuropsychological and C-IADL performance from baseline (pre-treatment) to end of treatment. Participant P1 self-reported a slight improvement in everyday executive functioning on the BRIEF but demonstrated a slight decline in performance on neuropsychological measures and slightly worse performance on the WCPA. This may have confounded by her report of fatigue, particularly in the follow-up visit. In contrast, participants P2 and P3 both demonstrated an improvement in select neuropsychological tests and the WCPA. Thus, clinical response to the intervention was variable but suggested that combining CCT and MST warrants further study.

To the best of our knowledge, this pilot study is among the first to evaluate a remotely delivered intervention specifically targeting executive functions in chronic stroke that combines CCT and MST. While the use of computerized cognitive training or metacognitive strategy training alone has been well-researched, we show here that the two approaches can be combined in a feasible and satisfactory way. The inclusion of metacognitive strategy training in addition to computerized cognitive training may help transfer gains to real-world C-IADLs [33]. In our three cases, we found that all three were able to perceive links between CCT exercises, MST exercises, and real-world C-IADLs personalized to them (e.g., functioning at work, leisure, or other activities).

We articulate in Table 1 the questions raised by our study and the next steps of intervention development. Specifically, future work should evaluate whether and how CCT + MST interventions can be made optimally accessible, for example by using web-based, app-based, and/or digital training exercises and homework. Whether the intervention remains feasible, acceptable, and shows a signal of clinical efficacy and transfer of training gains can be determined by a larger pilot randomized controlled trial. Importance should be placed on understanding contextual factors—aspects of patient history or clinical presentation, as well as stroke characteristics (lesion size, structural and functional connectivity)—and how they may predict response to the intervention. Future work should also quantitatively evaluate participants’ prior technology use or comfort with technology to guide personalized adaptation of the telehealth components of the intervention. It remains an open question to what extent in-person sessions are necessary or whether a CCT + MST intervention can be delivered fully remotely.

## 5. Limitations

This study had only three participants and did not have a control group or blinding, which limits any conclusions about clinical efficacy. Information on acceptability and feasibility should be considered preliminary. A follow-up pilot randomized controlled trial is necessary to evaluate feasibility and satisfaction more robustly and to provide initial evidence of clinical efficacy. Whether a combined CCT + MST intervention leads to greater clinical gains, and transfer to everyday C-IADLs, than CCT or MST alone cannot be determined by our study. A larger sample is also necessary to determine contextual factors that moderate feasibility, satisfaction, and potential efficacy. This is important in determining which individuals are most appropriate for a CCT + MST intervention. This study was conducted prior to the COVID-19 pandemic, and thus it is unclear how individuals’ comfort with and interaction with the technology-based aspects of the intervention may have changed with greater ubiquity of telehealth services.

## 6. Conclusions

Initial pilot testing of a remotely delivered executive function intervention that combines computerized cognitive training with metacognitive strategy training suggests that it may be satisfactory and feasible in chronic stroke. Three pilot participants found the treatment to be highly satisfactory, logical, and were expecting of change, and completed a high percentage of sessions. Feasibility and efficacy should be further evaluated in larger trials with control groups and assessors blind to group assignment.

## Figures and Tables

**Table 1 ijerph-20-05714-t001:** Component of the intervention development process that were incorporated into the current study. CCT = computerized cognitive training; MST = metacognitive strategy training.

Action	Steps Taken in the Current Study	Key Findings	Next Steps and Questions Raised
Plan the development process	Identify and assess problem; Ask if intervention is needed; Draw on published interventions.	There is a need for efficacious and accessible interventions that target post-stroke executive dysfunction.	--
Involve stakeholders	Collected feedback on acceptability and engagement from patients.	Three pilot participants found the intervention to be acceptable and engaging.	Will a larger sample of participants also show similar acceptability and engagement? What aspects of the intervention can be modified to further enhance engagement and accessibility (e.g., web- or app-based CCT, use of electronic homework exercises)?
Establish team decision-making	Include individuals with relevant expertise.	Key team members included individuals with expertise in neuropsychology, cognitive rehabilitation, occupational therapy, physiatry, neuroscience, and scalable intervention development.	--
Review published literature	Review published research evidence to identify existing interventions and understand evidence base.	Rehacom and the Multicontext approach selected as the CCT and MST approaches based on evidence base.	As research and technology progresses, will Rehacom remain the preferred CCT approach? What is the optimal set and sequence of training exercises?
Draw on existing theories	Identify theories or frameworks to inform intervention.	Theories of brain plasticity, learning, strategy acquisition and application, and generalization.	--
Collect data	Use quantitative measures and collect qualitative information on cognitive strategies learned and used.	CSQ-8, CEQ, and treatment session completion percentage used to assess satisfaction, credibility, expectancy, and feasibility.	Consider further use of qualitative/mixed methods to guide refinement. Collect data in a larger sample of stroke survivors.
Understand context	Understand the context in which the intervention will be implemented.	Preliminary evidence of feasibility and acceptability in chronic stroke.	Can CCT + MST intervention be used in acute/subacute stroke? What aspects of the person and of the stroke and its effects on brain connectivity may predict intervention response?
Pay attention to future implementation of the intervention in the real world	Understand facilitators and barriers to reaching the population and “scaling up”.	A CCT + MST intervention can be implemented remotely in participant homes.	Can CCT + MST demonstrate efficacy in a larger controlled trial? How can remote implementation be enhanced e.g., using electronic assessment and electronic homework exercises? How can generalization to individual patient goals/functional activities be further enhanced?
Refine intervention	Generate ideas about content, format, and delivery.	Pilot participants completed a high percentage of CCT and MST sessions.	As described above, consider refinements related to CCT exercises, MST homework exercises, and remote delivery.

**Table 2 ijerph-20-05714-t002:** Summary of intervention using the Template for Intervention Description and Replication (TIDieR) checklist. CCT = computerized cognitive training; MST = metacognitive strategy training.

TIDieR Item	Description
Brief Name	Combined Computerized Cognitive Training (CCT) and Metacognitive Strategy Training (MST)
Rationale	Executive dysfunction is disabling after stroke. Treatment options are limited. Combining MST and CCT may lead to greater transfer and generalization of treatment gains and may more strongly modulate neural circuits thought to underlie post-stroke executive dysfunction. Telehealth delivery may increase intervention accessibility.
Materials	Participants: Laptop computer, preloaded CCT software, workbook with homework exercises, information sheet for using technology Intervention Provider: Intervention manual, published training materials for MST [10].
Procedures	CCT: Exercises targeting attention, working memory, and executive functions. Sequence of exercises progress from training low-level attention to higher-order executive functions. Exercises adapt to performance. MST: Guided questioning before and after CCT exercises. Telehealth MST sessions with exercises using functional activities. Independent homework exercises involving functional activities.
Intervention Provider	Clinical neuropsychologist. Training and consultation by an expert in the Multicontext approach (JT)
Modes of Delivery	Individual treatment. Hybrid in-person and telehealth, with most sessions conducted via telehealth (videoconference or telephone).
Locations	In provider’s office (academic medical center) and in participant’s home.
Dosage	Treatment sessions were completed over 5 weeks. 25 CCT sessions, 30 min in duration; 8 MST sessions, one hour in duration.
Tailoring	Training on technology/hardware personalized to participant ability and comfort. CCT exercises automatically adapt to individual performance. The number of select CCT exercises was tailored (e.g., participant who quickly progressed through low-level attention exercises was more quickly transitioned to higher level executive functioning exercises). CCT working memory exercise was consistent for all participants. After CCT and MST exercises, the clinician used guided questions to help the participant link the exercises to the individual’s personalized, everyday C-IADLs. The last MST session was tailored to individual participant goals and C-IADLs.
Modifications	No modifications were made during the study.
Adherence and Fidelity	Calculated as the percentage of CCT and MST sessions completed for each participant.

**Table 3 ijerph-20-05714-t003:** Participant scores on satisfaction, credibility, expectancy, and feasibility assessments. CCT = computerized cognitive training; CEQ = Credibility Expectancy Questionnaire; CSQ-8 = Client Satisfaction Questionnaire-8; MST = Metacognitive Strategy Training.

	P1	P2	P3
CSQ-8			
Mid-treatment	28	32	30
End-treatment	27	32	32
CEQ-Credibility			
Baseline	8	9	8.3
Mid-treatment	7	9	8.7
CEQ-Expectancy			
Baseline	80	100	90
Mid-treatment	70	100	70
CCT Sessions Completed	21/25 (84%)	25/25 (100%)	24/25 (96%)
MST Coaching Sessions Completed	8/8 (100%)	8/8 (100%)	8/8 (100%)

**Table 4 ijerph-20-05714-t004:** Performance on the Weekly Calendar Planning Activity (WCPA) at baseline and after completing the intervention in three pilot participants.

WCPA	P1	P2	P3
Number of Correct Responses			
Baseline	9	10	12
End-treatment	10	13	15
Change	+1	+3	+3
Strategies Used			
Baseline	12	6	5
End-treatment	4	9	9
Change	−8	+3	+3
Rules Followed			
Baseline	2	2	3
End-treatment	1	4	5
Change	+1	+2	+2

**Table 5 ijerph-20-05714-t005:** Scores on neuropsychological tests that are demographically adjusted based on available clinical normative data.

Neuropsychological Measure (Demographically Corrected z-Score)	P1	P2	P3
Digit Span			
Baseline	−1.33	−0.67	0.67
End-treatment	−1	−0.67	2 *
Symbol Span			
Baseline	0	−1.33	−0.67
End-treatment	−1 *	−1.33	0.33 *
Symbol Digit Modalities Test			
Baseline	−1.87	−3.41	−1.24
End-treatment	−2.23	−3.41	−1.55
Trail Making Test-A			
Baseline	−0.61	−0.46	−0.09
End-treatment	−1.95 *	0.58 *	−0.15
Trail Making Test-B			
Baseline	−5.58	−2.95	0.98
End-treatment	−5.35	−1.65	1.07
Stroop Word Reading			
Baseline	−1.3	−2.4	−1.5
End-treatment	−2	−1.8	1 *
Stroop Color Naming			
Baseline	−2.1	−2.7	−0.5
End-treatment	−2.9	−2.3	−0.9
Stroop Color-Word			
Baseline	−2	−2.6	−0.7
End-treatment	−1.7	−0.7 *	−0.6
Paced Auditory Serial Addition Test (2 s)			
Baseline	−2.72	−2.31	0.8
End-treatment	n/a	−2.1	0.8
Paced Auditory Serial Addition Test (3 s)			
Baseline	−2.49	−2.39	0.3
End-treatment	n/a	−1.02 *	0.72
BRIEF Behavior Regulation Index			
Baseline	0.33	−1	−0.6
End-treatment	0.6	−1.9	1.1 *
BRIEF Metacognitive Index			
Baseline	−1.1	−0.7	−1.4
End-treatment	−0.9	−1.4	−0.1 *
BRIEF Gen Executive Composite			
Baseline	−0.5	−0.9	−1.2
End-treatment	−0.3	−1.7	0.5 *

Note: All scores are reported as z-scores with a mean of 0 and standard deviation of 1. Following clinical conventions, we considered a change in 1 standard deviation or more to be clinically significant, which is denoted with an asterisk.

## Data Availability

Not applicable.

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
