# Peer review of "Feasibility and Acceptability of a Remotely Delivered Executive Function Intervention That Combines Computerized Cognitive Training and Metacognitive Strategy Training in Chronic Stroke"

_ijerph, 2023, doi:10.3390/ijerph20095714_

Round 1

Reviewer 1 Report

This manuscript describes the development of a remotely-delivered intervention that combines computerized cognitive training and metacognitive strategy training. This research is compelling, important, and will be of interest to a broad audience. This manuscript could be enhanced by clearer and simpler presentation of results, expansion on feasibility of the technology aspect of the intervention, and grounding in an intervention development framework.

The introduction of this manuscript is strong. CCT and MST are both complex interventions, and each is described clearly, accurately, and thoroughly. There is also a strong justification for attempting this combination of intervention elements. The aims are appropriate for an intervention development study and the design is reasonable. There are a few methodological questions:

1)      Why did the authors choose to evaluate satisfaction and credibility at the mid-point of the study? Was this to determine whether there were any feasibility flags that could be resolved along the way? If so, this process could be explained, along with any considerations the authors made when evaluating the mid-point data. Otherwise, it’s not clear what this adds.

2)      The Credibility and Expectancy Questionnaire is not well-described. I don’t understand what is meant by “we evaluated expectancy by the fourth item response” when three questions are listed in the description.

3)      A critical piece that is missing from the methods is a description of training needs related to the technology component of the intervention, and how feasibility of the remote delivery was evaluated specifically. Previous studies of remote delivery models of cognitive interventions for adults with stoke have outlined the importance of clear, structured training that is personalized to each participant. The authors reported that participants used a device provided by the study team and that some training took place at the start and mid-point of the study. But the details of this training are unclear. Did the authors collect any data about previous technology use or comfort with technology that can guide personalized training as part of a scaled-up intervention protocol?

4)      Are the authors using an intervention development framework to guide this line of research? Presenting a framework will strengthen the argument for this study design and sample size and enable readers to understand why this study is valuable and where the research is heading.

There are several areas of the results that are difficult to follow or interpret, and some structural changes could make the results much clearer:

1)      Considering that there were only 3 participants in this study, the authors may consider listing the scores for each of the feasibility measures and the Calendar Planning Activity. Using summary statistics (medians and ranges) makes results difficult to interpret. With only 3 participants, listing the scores will be clearer and more meaningful. For the Calendar Planning Activity, it would be easier to follow if only change scores were presented, rather than pre- and post-scores for each participant. This is hard to follow as is.

2)      In lieu of multiple bar charts, a table may be a more efficient and effective way to present pre- and post- scores. This will allow readers to see the full picture of acceptance and feasibility all at once for easier synthesis.

3)      There is too much detail about individual neuropsych performance. If the focus of this study is intervention development and feasibility, that should be the focus of the results. Perhaps a one paragraph overview to summarize changes in neuropsych performance (using effect sizes or comparing change to MCID) would be more interesting and meaningful. As presented, the results include too many extraneous details that do not seem relevant to this intervention development manuscript. Focus instead on whether change was clinically meaningful to justify whether this intervention is worth exploring further in larger studies.

The discussion likely needs some restructuring based on the comments above. But one important note on the discussion – the authors should not claim that participants qualitatively described benefit without providing data to support this in the results.

Finally, the limitations section bears further elaboration. While this study design is appropriate for initial intervention development, it is important to discuss here what we can and cannot learn or explicate from this study.

Author Response

Thank you to the reviewers for their thoughtful and helpful feedback on our manuscript. We appreciate the positive feedback regarding the importance of the research and its interest for a broad audience. Below we detail our revisions to the manuscript based on reviewer feedback. We hope that with these revisions, the manuscript will be suitable for publication in the International Journal of Environmental Research and Public Health.

Reviewer 1

1. Why did the authors choose to evaluate satisfaction and credibility at the mid-point of the study? Was this to determine whether there were any feasibility flags that could be resolved along the way? If so, this process could be explained, along with any considerations the authors made when evaluating the mid-point data.

Thank you for suggesting this clarification. Knowing that treatment adherence and dropout can be an issue in complex interventions such as ours, we sought to measure satisfaction at midpoint as an additional datapoint to evaluate satisfaction with the initial intervention sessions when potential benefits may not have been apparent to subjects. Similarly, we evaluated credibility and expectancy at midpoint because we wanted to explore if subjects would continue to find the intervention credible with an expectancy of change. We have added text to the manuscript (pg. 7-8) to clarify our rationale.

Added text:

-“We also administered the CSQ-8 at the midpoint of the intervention to explore participant satisfaction with the initial portion of the treatment, particularly because treatment adherence/dropout can be a concern in complex interventions such as ours.”

-“We also administered the CEQ at the midpoint of the intervention to explore whether participants continued to find the intervention credible, with positive expectancy of change, as this could again inform the potential for adherence and dropout.

2. The Credibility and Expectancy Questionnaire is not well-described. I don’t understand what is meant by “we evaluated expectancy by the fourth item response” when three questions are listed in the description.

Thank you for pointing this out. We have rewritten the description of the Credibility and Expectancy Questionnaire to better describe the measure and how the scores are calculated.

Text added to page 8: “The CEQ (29) was administered to evaluate how believable and logical the participant perceives the treatment to be, as well as their expectancy of change. The first three items ask on a 9-point Likert-type rating scale, how logical the intervention is perceived to be, how successful the participant thinks the intervention will be in raising quality of functioning, and how confident the participant would be in recommending the intervention to a friend with similar problems. The CEQ Credibility score is calculated as the average of these three items. The CEQ Expectancy score is determined using a separate item which asks the participant to rate “how much improvement in your functioning do you think will occur” by the end of the intervention. This question is rated on a 0-100% scale in 10 percentage point increments. The CEQ scores at the beginning of treatment was the primary outcome of interest. We also administered the CEQ at the midpoint of the intervention to explore whether participants continued to find the intervention credible, with positive expectancy of change, as this could again inform the potential for adherence and dropout.”

3. A critical piece that is missing from the methods is a description of training needs related to the technology component of the intervention, and how feasibility of the remote delivery was evaluated specifically. Previous studies of remote delivery models of cognitive interventions for adults with stoke have outlined the importance of clear, structured training that is personalized to each participant. The authors reported that participants used a device provided by the study team and that some training took place at the start and mid-point of the study. But the details of this training are unclear. Did the authors collect any data about previous technology use or comfort with technology that can guide personalized training as part of a scaled-up intervention protocol?

We agree that this was an omission and we have revised the Methods (Procedure subsection, pg. 8-9) to specifically describe aspects of technology and technology training. Unfortunately, we did not formally collect quantitative data about prior technology use. We have added this to the Limitations section.

Added text to Methods: “2.4.2. Technology Use. As part of the initial (in-person) CCT session, participants completed training in the use of the loaned laptop computer and Rehacom software. The study clinician (AJ) demonstrated use of the hardware and software and then the participants had the opportunity to practice with corrective feedback as necessary. The extent of this training and practice was individualized for each participant. One participant required minimal training as she routinely used computers for her occupation. Another participant required more repetitive practice, and a third participant had assistance from a caregiver to help with both hardware and software. All participants (and a caregiver if present) were given an instruction sheet with step-by-step instructions on starting and shutting down the laptop computer and logging into the Rehacom program. Participants also had the clinician’s contact information for troubleshooting needs if necessary.”

Added text to Discussion (pg. 14): “Future work should also quantitatively evaluate participants’ prior technology use or comfort with technology to guide personalized adaptation of the telehealth components of the intervention. It remains an open question to what extent in-person sessions are necessary or if a CCT+MST intervention can be delivered fully remotely.”

4. Are the authors using an intervention development framework to guide this line of research? Presenting a framework will strengthen the argument for this study design and sample size and enable readers to understand why this study is valuable and where the research is heading.

Yes, we used the UK Medical Research Council framework (Skivington K, Matthews L, Simpson SA, Craig P, Baird J, Blazeby JM, et al. A new framework for developing and evaluating complex interventions: update of Medical Research Council guidance. BMJ. 2021 Sep 30;374(n2061):1–11; O’Cathain A, Croot L, Duncan E, Rousseau N, Sworn K, Turner KM, et al. Guidance on how to develop complex interventions to improve health and healthcare. BMJ Open. 2019 Aug;9(8):e029954). We have rewritten the manuscript to include this information, and added a table (Table 1) that highlights the steps we undertook under this framework and the important next steps/questions raised. We also include a table (Table 2) that demonstrates the components of the intervention using the TIDieR checklist approach (Template for Intervention Description and Replication).

Added text (page 3): To guide intervention development, we incorporated the UK Medical Research Council framework for developing and evaluating complex interventions (26) and specifically, consensus guidelines on how to develop complex interventions to improve health as articulated by O’Cathain et al. (27). Table 1 provides each of the components of O’Cathain’s framework for intervention development, how they were incorporated in the current study, key findings, and next steps/follow-up questions. Table 2 summarizes our intervention using the Template for Intervention Description and Replication (TIDieR) checklist.”

Table 1 added to manuscript:

Action

Steps Taken in the Current Study

Key Findings

Next Steps and Questions Raised

Plan the development process

Identify and assess problem; Ask if intervention is needed; Draw on published interventions.

There is a need for efficacious and accessible interventions that target post-stroke executive dysfunction.

--

Involve stakeholders

Collected feedback on acceptability and engagement from patients.

Three pilot participants found the intervention to be acceptable and engaging.

Will a larger sample of participants also show similar acceptability and engagement? What aspects of the intervention can be modified to further enhance engagement and accessibility (e.g, web- or app-based CCT, use of electronic homework exercises)?

Establish team decision-making

Include individuals with relevant expertise.

Key team members included individuals with expertise in neuropsychology, cognitive rehabilitation, occupational therapy, physiatry, neuroscience, and scalable intervention development.

--

Review published literature

Review published research evidence to identify existing interventions and understand evidence base.

Rehacom and the Multicontext approach selected as the CCT and MST approaches based on evidence base.

As research and technology progresses, will Rehacom remain the preferred CCT approach? What is the optimal set and sequence of training exercises?

Draw on existing theories

Identify theories or frameworks to inform intervention.

Theories of brain plasticity, learning, strategy acquisition and application, and generalization.

--

Collect data

Use quantitative measures and collect qualitative information on cognitive strategies learned and used.

CSQ-8, CEQ, and treatment session completion percentage used to assess satisfaction, credibility, expectancy, and feasibility.

Consider further use of qualitative/mixed methods to guide refinement. Collect data in a larger sample of stroke survivors.

Understand context

Understand the context in which the intervention will be implemented.

Preliminary evidence of feasibility and acceptability in chronic stroke.

Can CCT+MST intervention be used in acute/subacute stroke? What aspects of the person and of the stroke and its effects on brain connectivity may predict intervention response?

Pay attention to future implementation of the intervention in the real world

Understand facilitators and barriers to reaching the population and “scaling up.”

A CCT+MST intervention can be implemented remotely in participant homes.

Can CCT+MST demonstrate efficacy in a larger controlled trial? How can remote implementation be enhanced e.g., using electronic assessment and electronic homework exercises? How can generalization to individual patient goals/functional activities be further enhanced?

Refine intervention

Generate ideas about content, format, and delivery.

Pilot participants completed a high percentage of CCT and MST sessions.

As described above, consider refinements related to CCT exercises, MST homework exercises, and remote delivery. 

Table 2 added to manuscript:

TIDieR Item

Description

Brief Name

Combined Computerized Cognitive Training (CCT) and Metacognitive Strategy Training (MST)

Rationale

Executive dysfunction is disabling after stroke. Treatment options are limited. Combining MST and CCT may lead to greater transfer and generalization of treatment gains and may more strongly modulate neural circuits thought to underlie post-stroke executive dysfunction. Telehealth delivery may increase intervention accessibility.

Materials

Participants: Laptop computer, preloaded CCT software, workbook with homework exercises, information sheet for using technology

Intervention Provider: Intervention manual, published training materials for MST (10).

Procedures

CCT: Exercises targeting attention, working memory, and executive functions. Sequence of exercises progress from training low-level attention to higher-order executive functions. Exercises adapt to performance.

MST: Guided questioning before and after CCT exercises. Telehealth MST sessions with exercises using functional activities. Independent homework exercises involving functional activities.

Intervention Provider

Clinical neuropsychologist.  Training and consultation by an expert in the Multicontext approach (JT)

Modes of Delivery

Individual treatment. Hybrid in-person and telehealth, with most sessions conducted via telehealth (videoconference or telephone).

Locations

In provider’s office (academic medical center) and in participant’s home.

Dosage

Treatment sessions were completed over 5 weeks. 25 CCT sessions, 30 minutes in duration; 8 MST sessions, one hour in duration.

Tailoring

Training on technology/hardware personalized to participant ability and comfort. CCT exercises automatically adapt to individual performance. The number of select CCT exercises was tailored (e.g., participant who quickly progressed through low-level attention exercises was more quickly transitioned to higher level executive functioning exercises). CCT working memory exercise was consistent for all participants. After CCT and MST exercises, the clinician used guided questions to help the participant link the exercises to the individual’s personalized, everyday C-IADLs. The last MST session was tailored to individual participant goals and C-IADLs.

Modifications

No modifications were made during the study.

Adherence and Fidelity

Calculated as the percentage of CCT and MST sessions completed for each participant.

5. Considering that there were only 3 participants in this study, the authors may consider listing the scores for each of the feasibility measures and the Calendar Planning Activity. Using summary statistics (medians and ranges) makes results difficult to interpret. With only 3 participants, listing the scores will be clearer and more meaningful. For the Calendar Planning Activity, it would be easier to follow if only change scores were presented, rather than pre- and post-scores for each participant. This is hard to follow as is.

We have made these changes. For the Weekly Calendar Planning Activity, we added change scores, though also kept pre/post scores as this is valuable clinical information, particular for readers who are familiar with this assessment.

6. In lieu of multiple bar charts, a table may be a more efficient and effective way to present pre- and post- scores. This will allow readers to see the full picture of acceptance and feasibility all at once for easier synthesis.

Thank you, we have removed the figures and now report CSQ-8, CEQ, and WCPA scores in table format (Table 3 and Table 4).

7. There is too much detail about individual neuropsych performance. If the focus of this study is intervention development and feasibility, that should be the focus of the results. Perhaps a one paragraph overview to summarize changes in neuropsych performance (using effect sizes or comparing change to MCID) would be more interesting and meaningful. As presented, the results include too many extraneous details that do not seem relevant to this intervention development manuscript. Focus instead on whether change was clinically meaningful to justify whether this intervention is worth exploring further in larger studies.

Thank you, we agree with this recommendation. We have condensed the multiple figures of neuropsychological test performance into one table (Table 5) and we clarify that following clinical convention, we consider a change of 1 SD to be clinically meaningful. In the text of the Results, we have greatly shortened the description of neuropsychological test performance and report change in performance briefly for each subject.

8. The discussion likely needs some restructuring based on the comments above. But one important note on the discussion – the authors should not claim that participants qualitatively described benefit without providing data to support this in the results.

We have revised the Discussion according to the changes made throughout the manuscript. We have removed the portion of the discussion that mentioned qualitative benefit. We include the following text on the next steps of intervention refinement and evaluation:

“We articulate in Table 1 the questions raised by our study and the next steps of intervention development. Specifically, future work should evaluate whether and how CCT+MST interventions can be made optimally accessible, for example by using web-based, app-based, and/or digital training exercises and homework. Whether the intervention remains feasible, acceptable, and shows a signal of clinical efficacy and transfer of training gains can be determined by a larger pilot randomized controlled trial. Importance should be placed on understanding contextual factors – aspects of patient history or clinical presentation as well as stroke characteristics (lesion size, structural and functional connectivity) – and how that may predict response to the intervention. Future work should also quantitatively evaluate participants’ prior technology use or comfort with technology to guide personalized adaptation of the telehealth components of the intervention. It remains an open question to what extent in-person sessions are necessary or if a CCT+MST intervention can be delivered fully remotely.”

9. Finally, the limitations section bears further elaboration. While this study design is appropriate for initial intervention development, it is important to discuss here what we can and cannot learn or explicate from this study.

We have expanded on the Limitations section to clarify what can and cannot be learned and to describe important next steps: “This study had only three participants and did not have a control group or blinding, which limits any conclusions about clinical efficacy. Information on acceptability and feasibility should be considered preliminary. A follow-up pilot randomized controlled trial is necessary to evaluate feasibility and satisfaction more robustly and to provide initial evidence of clinical efficacy. Whether a combined CCT+MST intervention leads to greater clinical gains, and transfer to everyday C-IADLs, than CCT or MST alone cannot be determined by our study. A larger sample is also necessary to determine contextual factors that moderate feasibility, satisfaction, and potential efficacy. This is important in determining which individuals are most appropriate for a CCT+MST intervention. This study was conducted prior to the COVID-19 pandemic and thus it is unclear how individuals’ comfort with and interaction with the technology-based aspects of the intervention may have changed with greater ubiquity of telehealth services.”

Reviewer 2 Report

The paper provides an interesting remote execution of CCT and MST in Chronic Stroke patients.

The paper requires some editing review for minor typos (e.g., _Participants) and check also the style of references e.g. (1) vs. [1].

In section 2.3.2 it is not quite clear for the reader which is the 4th item response.

In Figure 3 the "WCPA Percent Accuracy" appears twice.

The authors recognise the fact that the limited number of patients in the trial of the new method limits the ability to generalize conclusions. Despite this fact it is still interesting to publish the early indications of a full scale study.

It would be interesting for the reader to have an indication of how these early results compare with state of the art methods (e.g., Neurophysical Assessment).

Author Response

Thank you to the reviewers for their thoughtful and helpful feedback on our manuscript. We appreciate the positive feedback regarding the importance of the research and its interest for a broad audience. Below we detail our revisions to the manuscript based on reviewer feedback. We hope that with these revisions, the manuscript will be suitable for publication in the International Journal of Environmental Research and Public Health.

The paper requires some editing review for minor typos (e.g., _Participants) and check also the style of references e.g. (1) vs. [1].

Thank you, we have proofread to correct typos. We have changed the reference format to IJERPH with square brackets.

In section 2.3.2 it is not quite clear for the reader which is the 4th item response.

We have elaborated and clarified the description of the CEQ in section 2.3.2 (see response to Reviewer 1 above). Specifically, we have clarified the 4th item response, which refers to: “The CEQ Expectancy score is determined using a separate item which asks the participant to rate “how much improvement in your functioning do you think will occur” by the end of the intervention.”

In Figure 3 the "WCPA Percent Accuracy" appears twice.

Per Reviewer 1’s feedback, we have removed figures and replaced them with tables.

The authors recognise the fact that the limited number of patients in the trial of the new method limits the ability to generalize conclusions. Despite this fact it is still interesting to publish the early indications of a full scale study.

Thank you for this positive comment.

It would be interesting for the reader to have an indication of how these early results compare with state of the art methods (e.g., Neurophysical Assessment).

Per suggestions from Reviewer 1, and because of the small sample in which we cannot draw conclusions on efficacy or clinical change, we have decided to de-emphasize discussion of pre/post change in neuropsychological assessment scores. We note more broadly that our findings suggest that a combined CCT+MST intervention such as ours warrants further study. We do note that the feasibility of this primarily remote intervention is similar to prior studies that have shown high feasibility using telerehabilitation for motor and cognitive deficits after stroke.

Round 2

Reviewer 1 Report

The authors have thoroughly addressed my comments regarding the clarity and focus of the methods, results, and limitations. The two intervention development tables are strong additions to this paper that clearly situate this study in the larger program of research, and highlight critical next steps. My only remaining suggestion is to consider removing the second and third paragraphs of the disussion (lines 382-402) because these concepts were already described in the introduction and methods and do not add much to the discussion.